# Less is more: Developing an approach for assessing clustering at the lower administrative boundaries that increases the yield of active screening for leprosy in Bihar, India

**Nimer Ortuño-Gutiérrez** [1‡]*, **Pin-Wei Shih** [2‡], **Aashish Wagh** [3],
**Shivakumar Mugudalabetta** [3], **Bijoy Pandey** [4], **Bouke C. de Jong** [5], **Jan Hendrik Richardus** [6],
**Epco Hasker** [5]

**1** Damien Foundation, Brussels, Belgium, **2** University Charité-Universitätsmedizin Berlin, Berlin, Germany,
**3** Damien Foundation India Trust, Bihar, India, **4** State Leprosy Eradication Programme, Bihar, India,
**5** Institute of Tropical Medicine, Antwerp, Belgium, **6** Department of Public Health, Erasmus MC, University
Medical Center Rotterdam, Rotterdam, The Netherlands

‡ These authors share first authorship on this work.
* Nimer.OrtunoGutierrez@damiaanactie.be

journal.pntd.0010764

Mexico Federico Gomez, MEXICO

**Data Availability Statement:** The data supporting
the findings of this publication are retained at the

## Abstract

### Background

In India, leprosy clusters at hamlet level but detailed information is lacking. We aim to iden-
tify high-incidence hamlets to be targeted for active screening and post-exposure
prophylaxis.

### Methodology

We paid home visits to a cohort of leprosy patients registered between April 1st, 2020, and
March 31st, 2022. Patients were interviewed and household members were screened for
leprosy. We used an open-source app(ODK) to collect data on patients' mobility, screening
results of household members, and geographic coordinates of their households. Clustering
was analysed with Kulldorff's spatial scan statistic(SaTScan). Outlines of hamlets and popu-
lation estimates were obtained through an open-source high-resolution population density
map(https://data.humdata.org), using kernel density estimation in QGIS, an open-source
software.

### Results

We enrolled 169 patients and screened 1,044 household contacts in Bisfi and Benipatti
blocks of Bihar. Median number of years of residing in the village was 17, interquartile range
(IQR)12-30. There were 11 new leprosy cases among 658 household contacts examined
(167 per 10,000), of which seven had paucibacillary leprosy, one was a child under 14
years, and none had visible disabilities. We identified 739 hamlets with a total population of
802,788(median 163, IQR 65–774). There were five high incidence clusters including 12%

Institute of Tropical Medicine, Antwerp, and will not be made openly accessible due to ethical and privacy concerns. Data can however be made available after approval of a motivated and written request to the Institute of Tropical Medicine at ITMresearchdataaccess@itg.be.

**Funding:** This work was supported by Damien Foundation Belgium grant for operational research to NO-G, AW,and SM. The funders had no role in study design, data collection and analysis, decision to publish, or preparation of the manuscript.

**Competing interests:** I have read the journal's policy and the authors of this manuscript have the following competing interests: Nimer Ortuño-Gutiérrez is employee of Damien Foundation. Aashish Wagh and Shivakumar Mugudalabetta are employees of Damien Foundation India Trust.

of the population and 46%(78/169) of the leprosy cases. One highly significant cluster with a relative risk (RR) of 4.7($p<0.0001$) included 32 hamlets and 27 cases in 33,609 population. A second highly significant cluster included 32 hamlets and 24 cases in 33,809 population with a RR of 4.1($p<0.001$). The third highly significant cluster included 16 hamlets and 17 cases in 19,659 population with a RR of 4.8($p<0.001$). High-risk clusters still need to be screened door-to-door.

## Conclusions

We found a high yield of active household contact screening. Our tools for identifying high-incidence hamlets appear effective. Focusing labour-intensive interventions such as door-to-door screening on such hamlets could increase efficiency.

### Author summary

India is the highest-burden country in the world where leprosy is known to cluster in hamlets. As no geographical data about hamlets is available, we develop a system to outline them using a high-resolution population density map. Then, using the household coordinates of 169 new leprosy cases enrolled in two hyperendemic blocks of Bihar, we screened household contacts and assessed clustering at hamlet level. The patients interviewed had lived in their current villages for a median of seventeen years at the moment of the survey. We found 11 new cases among 658 contacts examined equivalent to 16.7 per 1,000 population. There were three statistically significant clusters among five at the hamlet level and three including 78 cases in 98,623 population. Our results can be used to guide targeted and more efficient active case finding and post-exposure prophylaxis.

## Introduction

Leprosy is a complex infectious disease well known for centuries. *M. leprae* the pathogen that provokes leprosy, [1] multiplies in nerve and skin cells provoking skin lesions and neurological symptoms that vary from sensory loss, muscle weakness, and complete palsy, that later cause mutilations and deformities if not properly managed. The life-long disability associated with disfigurement is the main cause of stigma and discrimination. [2] This aspect, associated with the lack of diagnostic tests, limited access to proper and early care, and a long incubation period that could last decades, hinder the control of leprosy. The World Health Organization (WHO) concluded that the highly efficacious treatment with multidrug therapy (MDT) was insufficient to reduce the over 200,000 new cases annually that are notified for over a decade and recommends scaling up the preventive measures. [3] Based on the pivotal trial on chemo-prophylaxis in leprosy (COLEP) in Bangladesh, the WHO recommends post-exposure pro-phylaxis with single-dose rifampicin (SDR-PEP) for healthy persons in contact with new leprosy cases. [4] Provision of SDR-PEP combined with active case finding and early care of leprosy is recognized as key for stopping transmission. Determining the population at risk for the provision of active case detection and SDR-PEP is key for planning programmatic imple-mentation. [5] Although the programmatic implementation of SDR-PEP was successful in seven endemic countries, [6] the selection of those at higher risk and determining the opera-tional areas for active case detection and SDR-PEP are lacking.

India is the country that notifies the largest number of new leprosy cases accounting for 60% worldwide and is recognized as one of the 23 WHO priority countries for leprosy control. [7] In 2019, Bihar was ranked fifth among 37 states of India, with 16,595 new leprosy cases reported, equivalent to an annual new case detection rate (ANCDR) of 1.31 cases per 10,000 population. Among these patients 1,694 (10.2%) were children and 458 (2.8%) had grade two disability (G2D), among the latter 15 were children. These indicators are higher compared to the national level in 2019, with an ANCDR of 0.81 per 10,000 population, 6.9% of children, and 2.4% G2D. [8] In recent years India implemented innovative active case-finding (ACF) strategies [9] and included SDR-PEP in the control strategy. These strategies require substantial support in terms of human and financial resources. Identifying people at high risk for enhancing effectiveness and rationalizing resources are key to sustaining these activities in the long run.

Leprosy is known to be distributed unevenly and to cluster [10,11], focusing on the lowest geographical unit at high risk is key in view of the limited resources available. [10]

In India, villages are comprised of small hamlets known as 'Tola', [11] that often share common sociodemographic characteristics such as caste or religion. [12] These hamlets are informal subdivisions within villages, and formal boundaries and population estimates are lacking. Most disease control programs, including the national leprosy eradication program (NLEP) record patients at the village level. Mapping leprosy patients at the household level and using GIS-based tools to outline hamlets has great potential in improving the targeting of control measures. [13]

In the present study, we aim to develop and pilot spatial methodologies to outline hamlets within villages and then identify clusters of hamlets with high leprosy incidence. As such, we aim to contribute to curbing transmission as targeted by the WHO Global Leprosy Strategy 2021–2030 [14] by providing useful information to the policymakers for efficient targeting of interventions such as ACF activities and SDR-PEP in leprosy endemic countries.

## Methods

### Ethics

Our study obtained ethical clearance from the Institutional Review Board (IRB) from the Institute of Tropical Medicine, Antwerp (Number 1182/17), the University of Antwerp (Number B300201733691), and the ethical board of the Krishna Institute of Medical Sciences, Andhra Pradesh, India (No number). All new leprosy patients and their household contacts that gave their consent were included in the study. For children participants, formal consent was obtained from the parent/guardian.

### Study design

This is a cross-sectional study, we recruited new leprosy cases diagnosed between April 1st, 2020 and March 31st, 2022 in two blocks of Madhubani District of Bihar State, India.

### Study setting

In 2021, the state of Bihar had an estimated population of 123,083,000 inhabitants with 1037 inhabitants per km$^2$, ranked third among 37 states according to population density. [15] Around 80% of the population lives in rural areas, with an average household size of 4.8 members. Children under 15 years represent 36% and the sex ratio population is 1,009 females per 1,000 males. Around 60% of households are nuclear. The head of households are women in 23%. Concerning religion, 86% belong to Hinduism and 14% to the Muslim religion. Only 9%

have water piped into the households, and 39% do not use any toilet facility. Most of the population lives from agriculture, with 39% of the population cultivating their land and 57% owning farm animals. [16]

Among the 38 districts of Bihar, we selected Madhubani which notifies around 800 new leprosy cases annually. Within Madhubani districts we selected the blocks of Benipatti and Bisfi, accounting for respectively 404,457 and 358,913 estimated inhabitants in 2021 distributed over 267 villages. [17] In the last decade, Benipatti and Bisfi had notified an annual average of 190 and 182 new leprosy cases per million inhabitants, among those child proportion and G2D proportion were 13% and 0.4% and 25% and 2% respectively for the two blocks. From 2012 to 2021, the average proportion of multibacillary (MB) and female cases was 39% and 48% in Benipatti compared to 36% and 54% in Bisfi.

The National Leprosy Eradication Programme (NLEP) in Bihar implements passive case finding at the Primary Health Care facilities where persons with skin lesions present themselves. In 2016, the NLEP started active leprosy case detection campaigns (LCDCs) where trained health staff with the involvement of Accredited Social Health Activists (ASHAs) conducted door-to-door leprosy screening in the households of villages ranked in leprosy high priority districts (defined as prevalence rate superior to one per 10,000 inhabitants in the last three years). [9] In Bihar, two rounds of the LCDCs were implemented. [18]

## Sample size

Out of 224 new leprosy patients reported in Bisfi and Benipatti over the study period (April 1st, 2020-March 31st, 2022), we aimed to enroll approximately 200. This would allow us to estimate any 50% proportion with a precision of ±7%.

## Inclusion and exclusion criteria

We included all new leprosy patients diagnosed between April 1st, 2020 and March 31st, 2022 that accepted to be enrolled. We also included consenting permanent household members, including non-permanent, for screening for leprosy.

## Data collection

We obtained a line listing of new leprosy patients diagnosed in both blocks during the study period. The index cases were visited following the order of registration in the leprosy register, if absent the next index case was visited. We developed one app for data collection using the open-source Open Data Kit (ODK) forms. This allowed collection of information about household contact screening, demographic data, results of screening, and household geographic coordinates. A second ODK app included a questionnaire about the mobility of leprosy cases.

## Data analysis

Among new leprosy patients, we recorded the demographic and leprosy characteristics. We also recorded all permanent household members present at the visit and whether or not they were diagnosed with leprosy.

For outlining tolas and estimating their populations, we used population estimates from Humanitarian Data Exchange (HDX) which is an open-source platform for data sharing across crises and organizations managed by the United Nations Office for the Coordination of Humanitarian Affairs (UNOCHA). These population estimates are presented in raster files in Geotiff format, each raster point containing a value representing the estimated number of

people. For India and Pakistan high resolution, Geotiff files are available with a pixel size of approximately 30 x 30 meters [19].

Then, using Quantum Geographic Information System (QGIS) 3, we downloaded from OpenStreetMap the outlines of Bisfi and Benipatti Blocks. [20] We overlaid them with the corresponding population layer ('poplation_20_lon_80_general-v1.5') downloaded from HDX. [19] We used the outlines of the two blocks as mask layer to clip the corresponding part of the population raster.

Next, we transformed this clip of the population raster file to vector points with UTM 45N as coordinate reference system (CRS) using the 'Raster pixels to points' module in the Toolbox of QGIS. The shapefile created has only one field, 'VALUE', which we reset to 1 by dividing it by itself. We then created a heatmap based on kernel density estimation with a radius of 100 meters (including 3–4 pixels) and a raster size of 10 meters. Raster size was based on the assumption that the total population of the two blocks of approximately 800,000 is spread out over a surface area of approximately 450 km$^2$ of which approximately 20% is built-up area. Thus in the built-up area, there is approximately 1 person per 100 m$^2$ on average. This heatmap was converted to a raster, using the 'Raster calculator' from the QGIS Toolbox, selecting pixels with a value > 1.

This raster was again converted to a vector layer using the 'Polygonize (raster to vector)' function from the 'raster' menu of QGIS. From this vector layer, we removed all records with a value of zero. These were again overlaid with the clipped population data ('population_20_-lon_80_general-v1.5'). We then applied the 'zonal statistics' option from the QGIS Toolbox to obtain population estimates for each hamlet. We removed hamlets with population estimates of less than 20. These we exported with CRS UTM 45N and added the row number as a unique ID variable for each hamlet.

As a next step, we plotted the household coordinates of all leprosy patients enrolled and used the 'count points in polygons' function in QGIS to determine the number of cases in each hamlet. We added the centroid of each hamlet making use of the 'geometry tools' in QGIS. Finally, we exported our map layer with hamlet ID, population, number of cases, and X and Y coordinates of the centroid to a CSV file that was used in a SaTScan analysis.

To identify clusters of high leprosy risk we used SaTScan v10 to calculate Kulldorff's spatial scan statistic, maximum cluster size was set to 5%, based on the assumption that clusters of more than 5% of the total population, i.e. > 40,000 are too large to conduct efficient interventions. As spatial window size, we opted for elliptic clusters and a minimum of 5 patients per cluster. [21]

## Results

### Characteristics of leprosy index cases

During the study period (April 1$^{st,}$ 2020 to March 31$^{st,}$ 2022), 224 new leprosy patients were notified in the two study blocks, of which we enrolled 169(76%). These include 78/130 (60%) from Benipatti and 91/94(97%) from Bisfi. Out of those 72 (43%) were detected through active case finding (ACF), 40 of those through the active screening campaigns by ASHAs.

Age and gender distribution were comparable between the two blocks but the proportion of MB cases was highest in Benipatti (58% vs. 39%). Grade 2 disabilities (G2D) at time of diagnosis were more common in Benipatti (15%) than in Bisfi (5%). (Table 1)

### Patterns of clustering of new leprosy cases

We identified 739 Tolas with a total population of 802,788. The median population per tola was 163, IQR (65–774) with a maximum population size of 45,954 and minimum population

Table 1. Demographic characteristics of index cases enrolled.

| Block | Benipatti | (%) | Bisfi | (%) |
|---|---|---|---|---|
| Children | 12 | (15%) | 12 | (13%) |
| Female | 38 | (49%) | 50 | (55%) |
| MB | 31 | (58%) | 44 | (39%) |
| G2D | 9 | (15%) | 4 | (5%) |
| **Total** | **78** | **(100%)** | **91** | **(100%)** |

The participation rate was better in Bisfi, reasons for not being enrolled in the study were: temporary migration, migration of female cases after marriage, address cannot be traced, death, and mobility restrictions because of the Covid-19 pandemic.

size of 29. Fig 1 below shows the tola's outline with a EarthExplorer image as background for part of the study area.

Out of 169 cases included, 156 could be attributed to any of the hamlets, and 13 were outside the hamlets identified. Our SaTScan analysis identified five clusters with no overlap varying in size from 6,324 to 31,809 population and including 78 of 156 cases. Relative risk ranged from 4.14 till 5.1, three clusters were statistically significant. Out of an estimated total population of 802,788, we thus were able to select a population of 98,623 (12%) in which 46% of reported cases had occurred. Details are shown in Table 2 below.

In Fig 2. We display the high-risk clusters identified.

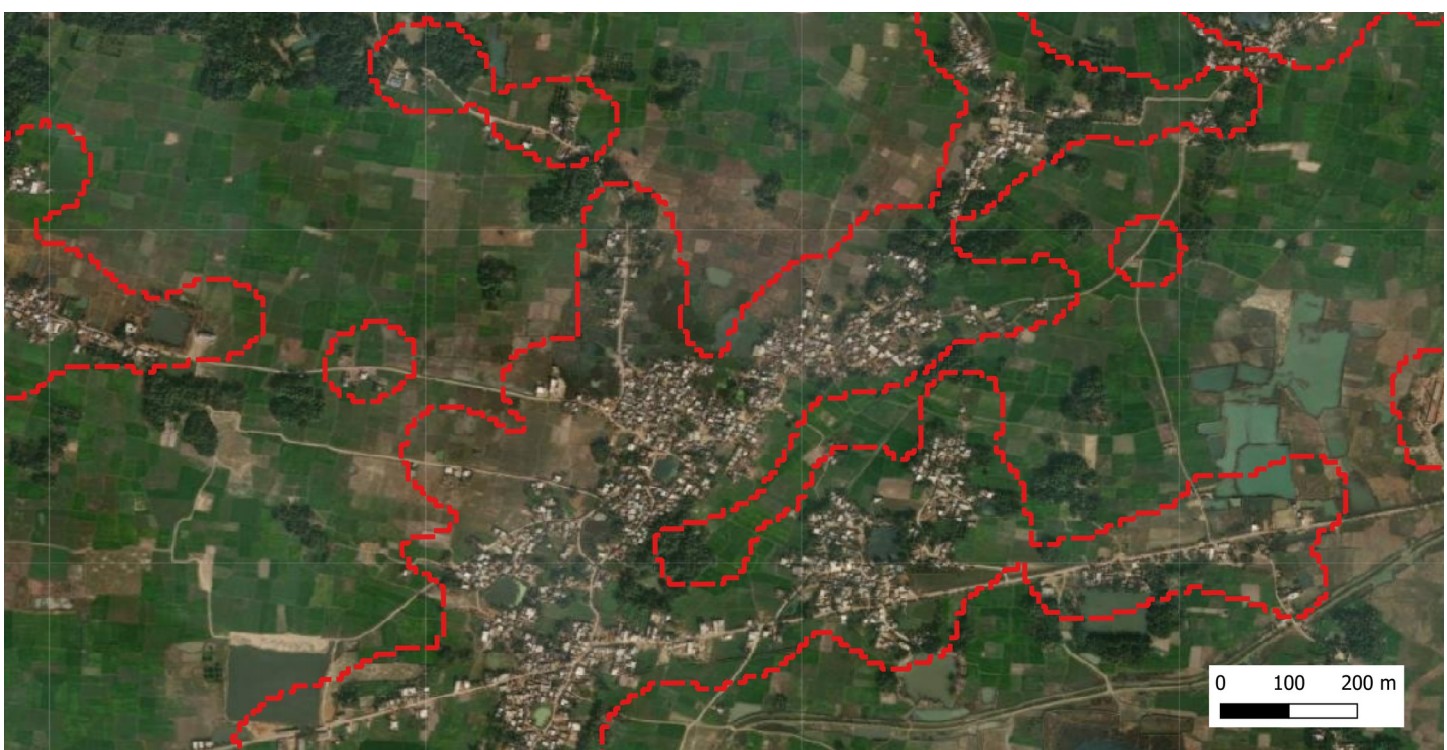

**Fig 1. Tola's outlined shown against EarthExplorer background.** We used United States Geological Survey (USGS) Map as a base layer for this map (http://www.usgs.gov).

Table 2. **Main clusters by Tolas in Bisfi and Benipatti, 2021–2022, India.**

| Cluster | Number of locations | Cases | Population | *RR | P–value |
|---|---|---|---|---|---|
| 1 | 32 | 27 | 33,609 | 4.7 | <0.0001 |
| 2 | 32 | 24 | 33,809 | 4.1 | 0.0006 |
| 3 | 16 | 17 | 19,659 | 4.8 | 0.001 |
| 4 | 9 | 5 | 5,222 | 5.1 | 0.939 |
| 5 | 10 | 5 | 6,324 | 4.2 | 0.993 |
| All | 99 | 78 | 98,623 | | |

*RR = Relative risk

## Mobility characteristics of leprosy index cases

Most patients had been living in the same village for many years. The median number of years that the leprosy index cases resided in the same village was 17 (IQR 12–35) and 18 (IQR 12–30) in Benipatti and Bisfi respectively. Only two leprosy index cases had been living in their village for ≤ one year and more than 80% had been living there for ≥ 11 years (Table 3).

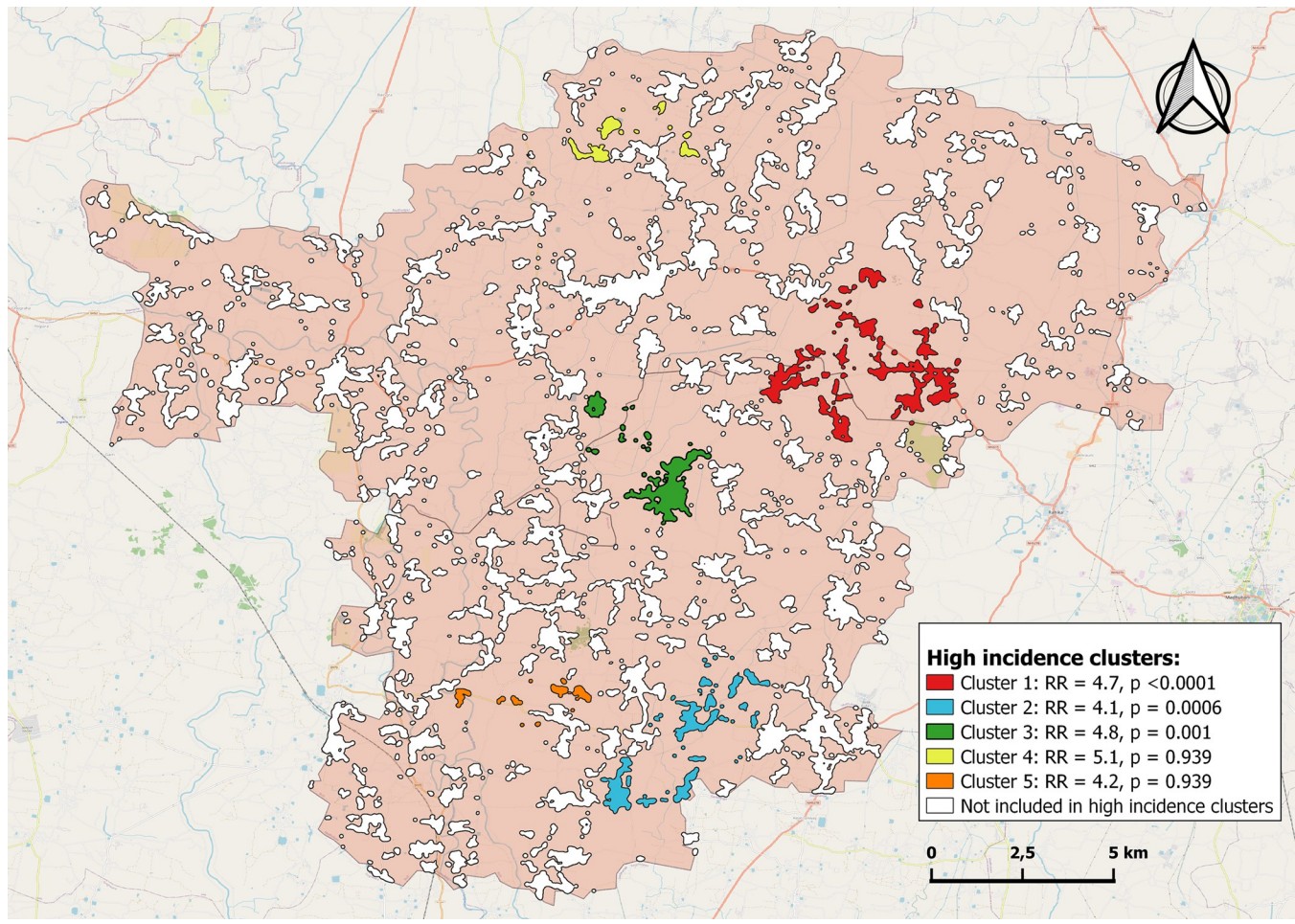

**Fig 2. High incidence clusters identified, Benipatti and Bisfi, India, 2020–2022.** We used OpenStreet Map as a base layer for this map (https://www.openstreetmap.org/#map=8/50.510/4.475).

**Table 3. Years of residence of leprosy index cases by block.**

| Block | Benipatti | | Bisfi | |
|---|---|---|---|---|
| **Years living in the village** | **n** | **(%)** | **n** | **(%)** |
| ≤1 year | 1 | (1%) | 1 | (1%) |
| 2–10 years | 14 | (18%) | 16 | (18%) |
| ≥11 years | 63 | (81%) | 74 | (81%) |
| **Total** | **78** | **(100%)** | **91** | **(100%)** |

## Demographic characteristics of household contacts and results of screening for leprosy

Index cases enrolled were part of 172 households, accounting for 1,044 contacts listed, from which 663 (63.5%) were present. The median household size was 6 in Benipatti (IQR 5–7) and 7 (IQR 6–8) in Bisfi. We examined 99% of household contacts present at the time of visit (Table 4). Among those, we found 11 (166 per 10,000) new leprosy cases belonging to nine households, and two households had two new leprosy cases. There were 7 PB (63%) cases, one child ≤ 14 years old (9%), and none had G2D.

## Discussion

We screened household contacts of 169 leprosy patients and identified 11 new cases among 663 persons screened, equivalent to a prevalence rate of 166 per 10,000. Making use of an innovative methodology we were able to outline 739 hamlets, part of the 267 villages in the two study blocks, and obtained population estimates for each. Plotting the geographic coordinates of leprosy index case households visited, we could identify five high incidence clusters of which three were statistically significant. Focussing on the high incidence clusters only would allow to select 12% of the population in which 46% of incident cases had occurred.

Our results illustrate once again that leprosy clusters at household level and that active screening of household contacts is highly efficient. Similar high yield of active screening of household contacts was reported from Comoros. [22] However, focusing only on household contacts will miss an important number of new leprosy cases as the increased risk of leprosy extends beyond households. [11] Using a GIS-based approach we were able to identify larger high-risk clusters beyond household level that would potentially benefit from measures such as active screening and post-exposure prophylaxis.

We also observed that in our study area mobility of leprosy patients is remarkably limited. The vast majority of patients interviewed (81%) had been living in their villages for more than

**Table 4. Demographic characteristics and results of screening for leprosy among household contacts.**

| Block | Benipatti | (%) | Bisfi | (%) |
|---|---|---|---|---|
| **Total listed** | **483** | **(100%)** | **561** | **(100%)** |
| Female | 222 | (46%) | 251 | (45%) |
| Children | 147 | (30%) | 186 | (33%) |
| Present at the visit | 330 | (68%) | 333 | (59%) |
| **Accepted examination** | **328** | **(99%)** | **330** | **(99%)** |
| Results of examination: | | | | |
| New case | 4 | (1.2%) | 7 | (2.1%) |
| Past leprosy or under treatment | 92 | (28%) | 97 | (29.4%) |
| No leprosy | 232 | (70.8%) | 226 | (68.5%) |

10 years. Taking into account also the long incubation period of leprosy, it may be worthwhile to consider not just current leprosy patients as index cases but also those detected in preceding years.

The methodology described allows to identify clusters of leprosy cases at a level below the village, which is the geographical unit most often recorded. It may be useful for other infectious diseases as well, such as visceral leishmaniasis and its sequel PKDL. [11,23] For leprosy, this approach contributes evidence to the Global Partnership for Zero Leprosy (GPZL) research question about the development of focused and adaptive sampling methods for efficient detection of local hot spots. [24] Other methods such as social network analysis can be further explored as addons in view of their usefulness observed in similar contexts. [25]

A potential limitation is a bias introduced by active case finding. If active case finding is focused on household contacts of leprosy patients or high incidence tolas, new cases arising close to previous cases are more likely to be detected. This effect may even have been exacerbated by the Covid-19 pandemic that made passive case finding services unavailable for prolonged periods. However, in our study, most cases were identified passively, and most actively detected cases were identified in a survey by ASHAs, who are present in all villages and whose screening campaigns target entire districts. [18] Furthermore, despite the fact that screening of household contacts is recommended, we still identified a fairly large hidden prevalence within the households of index cases.

The methodology described relies on mapping at household level of leprosy index cases. This is certainly possible with the technology currently available to the vast majority of health workers. Mobile smartphones with GIS software are universally used and open source software that allows mapping is widely available. Visiting the households of current and former leprosy patients is also highly valuable for the purpose of contact screening, as demonstrated again in this study.

## Conclusion

We have developed a method for outlining clusters of high leprosy incidence that warrants further exploration in other settings. The method is easy to apply and based on various open source software. It does require mapping at household level of leprosy patients which should no longer be a major hurdle with the tools currently available. Thus we will be able to focus preventive activities such as ACF and PEP where they are most needed.

## Acknowledgments

To the NLEP of the State of Bihar, the staff of Damien Foundation India Trust, ITM, and Erasmus University.

## Author Contributions

**Conceptualization:** Nimer Ortuño-Gutiérrez, Pin-Wei Shih, Aashish Wagh, Shivakumar Mugudalabetta, Bijoy Pandey, Bouke C. de Jong, Jan Hendrik Richardus, Epco Hasker.

**Data curation:** Nimer Ortuño-Gutiérrez, Pin-Wei Shih, Aashish Wagh, Epco Hasker.

**Formal analysis:** Nimer Ortuño-Gutiérrez, Pin-Wei Shih, Aashish Wagh, Jan Hendrik Richardus, Epco Hasker.

**Funding acquisition:** Nimer Ortuño-Gutiérrez, Aashish Wagh, Shivakumar Mugudalabetta, Bijoy Pandey, Jan Hendrik Richardus, Epco Hasker.

**Investigation:** Nimer Ortuño-Gutiérrez, Pin-Wei Shih, Aashish Wagh, Shivakumar Mugudalabetta, Bijoy Pandey, Bouke C. de Jong, Jan Hendrik Richardus, Epco Hasker.

**Methodology:** Nimer Ortuño-Gutiérrez, Pin-Wei Shih, Aashish Wagh, Bijoy Pandey, Bouke C. de Jong, Jan Hendrik Richardus, Epco Hasker.

**Project administration:** Nimer Ortuño-Gutiérrez, Aashish Wagh, Shivakumar Mugudalabetta.

**Resources:** Nimer Ortuño-Gutiérrez, Aashish Wagh, Epco Hasker.

**Software:** Nimer Ortuño-Gutiérrez, Pin-Wei Shih, Epco Hasker.

**Supervision:** Nimer Ortuño-Gutiérrez, Pin-Wei Shih, Aashish Wagh, Shivakumar Mugudalabetta, Bouke C. de Jong, Jan Hendrik Richardus, Epco Hasker.

**Validation:** Nimer Ortuño-Gutiérrez, Pin-Wei Shih, Aashish Wagh, Shivakumar Mugudalabetta, Bijoy Pandey, Bouke C. de Jong, Jan Hendrik Richardus, Epco Hasker.

**Visualization:** Nimer Ortuño-Gutiérrez, Pin-Wei Shih, Jan Hendrik Richardus, Epco Hasker.

**Writing – original draft:** Nimer Ortuño-Gutiérrez, Epco Hasker.

**Writing – review & editing:** Nimer Ortuño-Gutiérrez, Pin-Wei Shih, Aashish Wagh, Shivakumar Mugudalabetta, Bijoy Pandey, Bouke C. de Jong, Jan Hendrik Richardus, Epco Hasker.

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
