## [Decision Letter · Decision Letter 0]

23 Aug 2022

Dear Dr.  Ortuño-Gutiérrez,

We are pleased to inform you that your manuscript 'Less is more: developing an approach for assessing clustering at the lower administrative boundaries that increases the yield of active screening for leprosy in Bihar, India' has been provisionally accepted for publication in PLOS Neglected Tropical Diseases.

Best regards,

Carlos Franco-Paredes

Academic Editor

Ana LTO Nascimento

Section Editor

Reviewer's Responses to Questions

**Key Review Criteria Required for Acceptance?**

**Methods**

-Are the objectives of the study clearly articulated with a clear testable hypothesis stated?

-Is the study design appropriate to address the stated objectives?

-Is the population clearly described and appropriate for the hypothesis being tested?

-Is the sample size sufficient to ensure adequate power to address the hypothesis being tested?

-Were correct statistical analysis used to support conclusions?

-Are there concerns about ethical or regulatory requirements being met?

Reviewer #1: The Objectives are straightforward, with good study design.

The Methods are complex, but clearly described.

**Results**

-Does the analysis presented match the analysis plan?

-Are the results clearly and completely presented?

-Are the figures (Tables, Images) of sufficient quality for clarity?

Reviewer #1: The Results are displayed very clearly.

**Conclusions**

-Are the conclusions supported by the data presented?

-Are the limitations of analysis clearly described?

-Do the authors discuss how these data can be helpful to advance our understanding of the topic under study?

-Is public health relevance addressed?

Reviewer #1: The Conclusions are solidly based on the findings.

**Editorial and Data Presentation Modifications?**

Reviewer #1: My only reservation concerns the point of giving p values for statistical significance. The p values jump from being highly significant (0.001 or less), to being totally non-significant (p > 0.9). It seems that the p values are almost entirely dpendent on the population under study, and are of no practical use. The relative risk is the key result and even small groups of people with a high RR are worth knowing about.

**Summary and General Comments**

Reviewer #1: A very nice paper.

PLOS authors have the option to publish the peer review history of their article (what does this mean?). If published, this will include your full peer review and any attached files.

Reviewer #1: **Yes: **Paul Saunderson

---

## [Editor Report · Acceptance letter]

7 Sep 2022

Dear Dr Ortuño-Gutiérrez,

We are delighted to inform you that your manuscript, "Less is more: developing an approach for assessing clustering at the lower administrative boundaries that increases the yield of active screening for leprosy in Bihar, India," has been formally accepted for publication in PLOS Neglected Tropical Diseases.

Best regards,

Shaden Kamhawi

co-Editor-in-Chief

Paul Brindley

co-Editor-in-Chief
